Design of Chinese painting style classification model based on multi-layer aggregation CNN

Du Xiaofang 1
Cai Yangfeng 2 fcyang1256@outlook.com
1 School of Architecture and Art Design, Jiangxi Technical College of Manufacturing , Nanchang, Jiangxi , China
2 Faculty of Art and Design, Jiangxi Institute of Fashion Technology , Nanchang, Jiangxi , China
Asif Muhammad
Electronic publication date: 2024 Oct 8
Publication date: 2024
Volume: 10
Electronic Location ID: e2303
Received 2024 May 23; Accepted 2024 Aug 12
Copyright: © 2024 Du and Cai
Copyright year: 2024
Copyright holder: Du and Cai
License: This is an open access article distributed under the terms of the Creative Commons Attribution License, which permits unrestricted use, distribution, reproduction and adaptation in any medium and for any purpose provided that it is properly attributed. For attribution, the original author(s), title, publication source (PeerJ Computer Science) and either DOI or URL of the article must be cited.
License URL: https://creativecommons.org/licenses/by/4.0/

Keywords: TCP, Emotion mapping, CNN, Weighted activation, Cultural heritage

Funding: The authors received no funding for this work.

==============================
This study delves deeply into exploring the artistic value of traditional Chinese painting (TCP) and aims to bridge the gap between its fundamental characteristics and the realm of human emotions. To achieve this, a novel convolutional neural network (CNN)-based classification model for TCP emotions is proposed. By thoroughly analyzing the distinct emotional mapping relationships inherent in TCP, a comprehensive framework is developed. Notably, emotional feature regions are accurately extracted using image saliency, and a multi-layer aggregation recalibrated emotional feature module is seamlessly integrated into the CNN network structure. This integration strengthens the activation of features that significantly influence TCP emotions. Moreover, a method of multi-category weighted activation localization is skillfully employed to classify Chinese painting emotions within the CMYK (cyan magenta yellow black) color space. The empirical results convincingly demonstrate that our algorithm surpasses existing approaches such as VGG-19, GoogLeNet, ResNet-50, and WSCNet in the task of TCP emotion recognition, achieving an impressive accuracy of 92.36%, the largest error is 0.191. This improvement signifies the advancements made by our model in accurately capturing and understanding the emotional nuances within TCP. By outperforming previous methods, our research contributes to the convergence of multimedia technology and cultural education.

Introduction

Traditional Chinese painting (TCP) stands as a cherished artistic heritage within Chinese cultural tapestry. It is an art form that places a paramount emphasis on imagery, allowing the artist to imaginatively and creatively interpret the subjects depicted within the canvas. TCP truly represents an invaluable cultural treasure. Distinguished TCP masterpieces are the profound manifestations of a painter’s sincere and heartfelt expression (Yuehao, 2021).

A substantial divergence exists between the feature information derived from rudimentary visual data and the nuanced interpretation of human emotions. This distinction is recognized as the “emotional gap” (Datta, 2009). In recent years, the domain of computer vision has witnessed a burgeoning interest in semantic image classification, with an especially dynamic focus on the realm of image emotion classification, representing the zenith of semantic categorization. Our objective is to empower computers to employ suitable models in order to articulate the emotional responses that individuals experience when engaging with TCP artworks, ultimately classifying them in accordance with these nuanced emotions. In contrast to cognitive semantics, emotional semantics tend to be more subjective, susceptible to variations rooted in individuals’ cognitive levels, cultural backgrounds, and aesthetic standards, all of which influence the interpretation of emotional semantics. The formidable attributes of deep learning algorithms offer a potent avenue to bridge the emotional gap and facilitate the comprehension of emotions (Kumar, 2021).

Deep learning is a feature-based learning method. Through some simple and nonlinear model transformation, the original data becomes a higher level and more abstract expression. At present, there are a lot of researches on image emotion analysis. He & Zhang (2018) used binary classification results to assist in multi classification of natural image emotion. However, the data set needs two sets of emotional tags, namely dichotomous and multiclassification, and the workload is increased (He & Zhang, 2018). Kang, Shim & Yoon (2018) constructed a color matching emotional word data set, found the corresponding emotional words in the data set by creating color spectrum for western paintings, and predicted the emotion of paintings. Compared with photos and oil paintings, the lines of TCP are more obvious, and the unique artistic pursuit of artistic conception is emphasized. The above research methods for the emotion of natural images or western paintings are not fully applicable to the analysis of TCPs. The intersection of psychology and art is a rich field that explores how visual stimuli evoke emotional responses. The emotional classification of TCP can provide valuable insights into these processes. Modern psychology investigates how people perceive and process emotions. By applying these principles to TCP, we can better understand how different elements of these paintings—such as brushwork, composition, and color—elicit specific emotional responses from viewers. Furthermore, the therapeutic potential of art is well-documented in psychological practice. Classifying and understanding the emotions in TCP can aid in developing art therapy programs that utilize these paintings to evoke therapeutic emotional responses, supporting mental well-being and emotional health.

At present, there is very little research on the emotion of Chinese painting. Meanwhile, the research methods on the emotion of natural pictures or western paintings are not fully applicable to the analysis of Chinese painting, because it contains high-level semantic information of subjective abstraction, which is the main challenge of this article. Due to the imaginative attributes of TCP, which centers around “freehand brushwork”, we unavoidably need to think about the profound variables in TCP in our appreciation and exploration so seeing significant emotion level profound semantics can help us appreciate and concentrate on TCP which is also conducive to the spread of TCP art.

The research motivation of this article is to delve into the artistic value of TCP through the exploration of computer-based methods for analyzing emotions in TCP. It aims to bridge the gap between low-level image features and the high-level semantics of human emotion, providing tools for users’ appreciation and learning. Additionally, the study aims to promote digital preservation and cultural heritage management. The article focuses on the principles of feature extraction and architectural optimization in Convolutional neural networks (CNNs), tailored to the unique artistic characteristics of TCP. It investigates methods for emotion classification and visualization in TCP, aiming to validate that the proposed algorithm accurately identifies emotional areas within TCP. By doing so, it aims to preserve the emotional integrity of artistic expression, facilitating easier learning and appreciation of TCP techniques used by artists to convey their sentiments.

Related works

Analysis of the TCP characteristics

The feature extraction of TCP relies on image recognition, and visual attention has become a widely used component. However, the category semantic features are easily entangled with each other, while CNN can explicitly extract the category perception features, Cheng et al. (2023) introduced a method using class-specific dictionary learning to disentangle neural network outputs into category-aware features. Cheng et al. (2018) proposed discriminative CNNs (D-CNNs) to address challenges such as within-class diversity and between-class similarity. D-CNNs optimize a new discriminative objective function, incorporating metric learning to enhance feature discrimination (Cheng et al., 2018).

However, the current research on TCP focuses on the identification of the painter and the analysis of the style of the work, and the emotional research on TCP is very few. Jiachuan & Yuzhi (2018) took the characteristics of TCP brush movement into account and adopted CNN to extract the high-level semantic features of TCP depicting artistic objects, where SVM and improved embedded learning algorithm are introduced to realize the recognition of TCP painters. Zheng, Haoyue & Hongshan (2017) divided Chinese paintings into six categories according to emotion expression by using support vector machine after integrating Chinese painting features and color features extracted from CNN. Although CNN is effective in mining high-level emotional semantic features, it ignores a large number of low-level texture features in the training process, and the results are over fitted. Although various attention-based methods have been proposed and achieved relatively competitive results, it is observed that the semantic features of each class are likely to entangle with each other, and few works focus on explicitly extracting category-aware features so far. CNN is good at dealing with relatively complex feature extraction or classification tasks (Szegedy, Liu & Jia, 2015). Compared with visual geometry group net, the size of convolution layer of GoogLeNet is relatively fixed, and the semantic meaning of visualization is strong, which is conducive to the comparative observation of each layer of visualization, and helps to improve the network structure combined with the prior knowledge of human appreciation of TCP feelings (Szegedy, Liu & Jia, 2015; He & Zhang, 2018; Cheng et al., 2023). TCP elements such as stroke lines, depicting objects also reveal the author’s emotion, while GoogLeNet discards some learning methods of middle-level features and wastes a large number of emotional features expressed in TCP. At the same time, GoogLeNet auxiliary classifier plays a small role in Imagenet classification task and does not participate in the result prediction. Selvaraju et al. (2017) proposed gradient weighted class activation mapping (Grad CAM) technology. By calculating the gradient of the target category relative to the convolution layer, the class activation graph is generated, and the influence degree of different regions of the image on the results is visualized (Selvaraju et al., 2017). Olah, Mordvintsev & Schubert (2020) established visual CNN layer features to explain the reasons for network recognition of object types. The above visualization studies are all aimed at the field of target detection, while for the subjective and abstract high-level semantic information such as TCP emotion, it is more necessary to explore the relationship between the features extracted from deep network and the decision-making results and the reasons.

Image emotion analysis

Extensive research efforts have been dedicated to the automated analysis of emotions in images. In comparison to text data, images encompass a broader and more abstract spectrum of information. Some studies have endeavored to extract the fundamental features from images and subsequently analyze the emotional nuances by incorporating expertise from the field of human psychology or cognitive visual observation. For instance, Kang, Shim & Yoon (2018) constructed an emotional word dataset linked to color, devising a color spectrum specifically tailored to Western paintings. They ascertained that the emotional lexicon within the dataset can predict the emotional undercurrents present in the paintings (Kang, Shim & Yoon, 2018). Bo, Chen & Hui (2018) leveraged a weighted K-nearest neighbors (KNN) algorithm to anticipate subjective and elusive emotions inherent in abstract paintings, relying on the extraction of image color and texture attributes. Yao et al. (2015) achieved emotion recognition by isolating unit features inherent in facial expressions and establishing interconnections between them.

The advent of CNNs has significantly enriched various fields, particularly in image classification and recognition, demonstrating remarkable performance. Consequently, deep learning techniques have progressively penetrated the realm of image emotion analysis. You et al. (2015) introduced the Progressive CNN depth model based on CNN, and they assembled a substantial, manually annotated visual emotion dataset from Twitter. Jou & Chang (2016) addressed the challenge of image feature description and emotion classification through the exploration of multi-task deep learning. Campos, Jou & Giroi Nieto (2017) meticulously assessed the performance of CNN architectures layer by layer, fine-tuning their application for image emotion prediction, affirming the benefits of deep network learning in the recognition of emotional features within natural images.

However, due to the distinctive characteristics of TCP, marked by more conspicuous lines compared to photographs and Western paintings, which prioritize artistic interpretation over realism, these methodologies are ill-suited for TCP analysis (Yang et al., 2020).

Emotional characteristics in tcp

Color

In the final analysis, painting is the art of vision. Color can give people the most intuitive and strong psychological feeling. The same rules apply to the appreciation of TCP.

From the perspective of emotional space, emotional expression of the former may be more positive. Considering emotion from the perspective of color, Table 1 comprehensively reflects the set of emotion types to which the color may correspond.

Table 1 Color-emotion mapping in TCP.

Parameter	Details	
Base architecture	ResNet-50	
Pre-training dataset	ImageNet	
Number of network prediction nodes	8	
Initial learning rate	10–3	
Learning rate decay rate	0.9	
Optimizer	Adam	
Batch size	32	
Loss function	50	
Loss function	Cross-entropy loss	
Activation function	Activation function	
Dropout rate	0.5	
Regularization	L2 regularization	

Shape and texture

Color features are not the only information that affects the semantic features of a painting. Texture and shape can also have a significant impact on people’s psychology. The main shapes, lines and textures of the contents of a picture are different, and the emotions expressed are also different. Generally speaking, smoothness gives people a delicate feeling, while roughness gives people a sense of old age; The regular shape is comfortable, while the irregular shape is not. Tables 2 and 3 briefly describe the mapping relationship between shape and texture to emotion.

Table 2 Shape-emotion mapping in TCP.

Shape	Emotion	
Tidiness (regularity)	Comfortable	
Lack (irregularity)	Disgust, irritability	
Fusiform	Orthodox	
Square	Correct and concentrated	
Circular	Round and smooth relaxation	
Arc	Slack	
Curve	Dynamic	
Instable	Anxious	
Stable (symmetrical and uniform)	Calm	
Triangle	Mechanical, indifferent	

Table 3 Texture-emotion mapping in TCP.

Texture	Emotion	
Smooth	Delicate and relaxed	
Coarse stickiness	Old	
Soft	Warm and gentle	
Hard	Strong	

In addition to the more important color features and shape features, from other perspectives, such as image geometric features, gray histogram, frequency band changes, image understanding changes and so on, may help us calculate and analyze the emotional semantics of images. In addition, some unique intentions often correspond to specific meanings in TCP. For example, plum, orchid, bamboo and chrysanthemum generally represent high and clean character, which requires the connoisseur to have relevant knowledge background.

Tcp emotion classification model based on cnn

Feature region extraction

Because CNN requires the input image size to be 224 × 224 pixels, and the TCP works that happen to be square are very few. Because the painter infuses his inner feelings into the carefully designed and carved lines, shapes and other elements, in the natural image preprocessing, simple scaling or interception will seriously damage the emotional information of art paintings.

The main object of TCP is the area where the painter’s emotion is concentrated. The main body of the picture is generally concentrated in a region, which is usually bright and obvious, and the ink is more elaborately carved, which plays a role in highlighting the theme and creative intention. Therefore, the region of interest is extracted according to the image saliency and stroke complexity to obtain the region with rich emotional expression in TCP, which is helpful for the subsequent extraction of emotional features.

First, invert the color of the image. Let Ii=(ri,gi,bi)T be the three-dimensional color vector of the i -th pixel of image I. After color inversion, the pixel color vector Ci of image C can be calculated by Formula (1).

(1) Ci=(255−ri,255−gi,255−bi)T

Then, BMS is used to predict the saliency of the image. Attention saliency map D=BMS(C), where D is a binary image, and the pixel value of the region with high saliency is one and the pixel value of the background region is 0. The connected region with the same pixel value of 1 is taken as a significant region, search the minimum value xminand maximum value xmax of this region in the horizontal coordinate and the minimum value ymin and maximum value ymax in the vertical coordinate. Then the enclosing rectangle of the i-th significant region is represented in Formula (2):

(2) E(x,y)i=(xmin,ymin,xmax,ymax)

After the image is grayscale processed, the binary edge image of salient region is calculated by Formula (3):

(3) G(x,y)i=fb(∇sobel(fgray(E(x,y)i)))

where fgray(⋅) represents graying the surrounding rectangular region of significant area, ∇sobel means Sobel operator, fb(⋅) binary edge image through the sensitive threshold γ.

g(x,y), the pixel of G(x,y) is defined as shown in Formula (4):

(4) g(x,y)={1,|∇s(fgray (e(x,y)))|> γ0,|∇s(fgray (e(x,y)))|≤ γ

Then the external rectangle of the prominent area of TCP is shown in Formula (5):

(5) E(x,y)s=argmaxi∈[0,r]⁡∑j=0xmax−xmin⁡∑k=0ymax−ymin⁡g(j,k)i

Among them, n represents the number of circumscribed rectangles in prominent areas of TCP.

The side length of the prominent area of in square TCP is shown in Formula (6).

(6) l=δm(Iwidth,Iheight),δ∈(0,1]

where Iwidth and Iheight represent the width and height of image I respectively, and δ represents the clipping threshold. It can be seen from Formula (1) that in the highest complexity of the enclosing rectangle E(x,y), the coordinate of the upper left corner is (xmin,ymin), and the coordinate of the lower right corner is (xmax,ymax), then the coordinate of the center point of E(x,y) is shown in Formula (7).

(7) (xs,ys)=(⌈xmin+xmax2⌉  ,  ⌈ymin+ymax2⌉).

Optimized CNN structure

This article introduces an enhancement to the internal architecture of the ResNet-50. Capitalizing on the benefits of identity mapping, we introduce a novel Multi-Layer Aggregation Recalibration Emotional Feature Module. This module is founded on the concept of residual modules and is designed to facilitate the aggregation of feature information from multiple convolution layers within a single module. The amalgamated information is then seamlessly fed back into the output of the convolution layer. This augmentation results in the improvement of the convolutional neural network’s structure, achieved by recalibrating the activation intensity of various features of the module unit. This recalibration process serves to accentuate the feature activation responses associated with the emotional nuances found in TCP, thereby enhancing the network’s capacity to discern emotional attributes in TCP imagery. A visual representation of the feature recalibration module for deep aggregation is depicted in Fig. 1.

Figure 1 CNN structure optimization.

Assuming that the dimension of input X is h×b×c, and the output of the first convolution layer at the layer can be calculated as shown in Formula (8).

(8) A1=w1⊗X

where ⊗ is convolution operation. w1 is the kernel parameter. For clarity and simplicity, the formula ignores the offset term.

The output A of h×b×c dimension convolution layer is compressed into vector g(A) through global average pooling. The k-th element of vector g(A) is expressed as Formula (9):

(9) gk(A)=1h×b∑ib∑jhAijk

where Aijk is the i-th row and j-th column element of the k-th channel of convolution layer A. According to the above formula, the vectors after global average pooling operation of the three convolutional layers in the module are respectively expressed as g(A1), g(A2), g(A3).

The elements within the vector correspond to the feature information of each channel within the respective convolution layer. The scale of the feature information is contingent on the depth of the convolutional layer. In the shallower layers of the network, the convolutional kernel predominantly captures fine-grained details such as points, lines, and textures. This is primarily due to the larger resolution of the input or feature map. Consequently, the channels of the corresponding convolution layers encapsulate localized, intricate features within the image.

Conversely, as we delve into the deeper layers of the network, the feature map resolution diminishes, and the convolutional kernel evolves to capture a more extensive range of information. In this scenario, the channels of the corresponding convolution layers are more inclined to represent overarching, abstract features of the image. These features encompass a holistic depiction of objects and the emotional ambiance conveyed within the image. gk(A) calculates statistically the activation of A feature contained in the k-th channel of convolution layer A in the global scope of the image. The dimensions of vectors g(A1) and g(A2) are d, and the dimensions of g(A3) are c, But because d < c), it cannot be directly aggregated with the number of channels in the corresponding convolution layer. Therefore, g(A1) and g(A2) with the same dimension are firstly aggregated, then the vector dimension is raised to C dimension through the full connection layer, and then aggregated with g(A3). The calculation formula of the obtained vector e is as shown in Formula (10):

(10) e=φ(W1(g(A1)+g(A2)))+g(A3)

where W1∈  c×d is the parameter of full connection layer. φ(⋅) is the adaptive activation function, which is defined as shown in Formula (11):

(11) φ(x)=αfReLU(x)+βfELU(x)

α and β are network trainable parameters, fReLU(.) and fELU(.) are the ReLU activation function and ELU activation function respectively.

The network loss function is expressed as shown in Formula (12):

(12) L=−1U∑iU∑mnpimlogp^im

where U represents the number of input variables, n represents the classification category, pim is the m-th category tag, and p^im is the corresponding predicted tag.

Visual classification

To calculate the activation map Vm of m-th category, let qm represent the predicted value of m-th category before Softmax, set [q1,q2,⋯,qm,⋯,qn]=[0,0,⋯,1,⋯,0], that is, set the predicted value of the currently calculated category to 1. Calculate the gradient of qm with respect to channel A of the target convolution layer ∂qm∂Aijk, where Aijk represents the i-th row and j-th column elements of channel k. The gradient of channel k is globally averaged pooled, and the weight θkm of the channel affecting the m-th category is shown in Formula (13):

(13) θkm=1h×b∑ib∑jh∂qm∂Aijk

where h×b is the size of channel k. Next, the weighted sum of each channel is calculated to obtain the influence of features in the convolution layer on the m-th category, and ReLU is used to shield all negative activations, as shown in Formula (14):

(14) Vm=fReLu(∑kcθkmAk).

Finally, weighted aggregation of all categories of activation is performed to obtain the multi-category weighted activation location S, as shown in Formula (15):

(15) S=∑mnpmVm

where pm represents the predicted probability value of the image in the m-th category, and n represents the number of classification categories. This article defines the emotional category N = 8, and divides Chinese paintings into four categories: sad, lonely, pride, wanton, lively, cheerful, quiet and peaceful. To visualize each emotion clearly and without interfering with each other, the activation values were normalized, and different color single channels of CMYK color space were assigned to the eight emotion categories. In order to facilitate the display, the inverse colors of the four standard colors of the CMYK model, namely white, red, green and blue, were used to mark the Chinese paintings, respectively, to represent the positions of sad and lonely, pride and wanton, lively and cheerful, quiet and peaceful.

Experiment and analysis

Dataset

From the Internet (www.baidu.com), 800 TCPs from ancient to modern times were obtained. Based on the analysis of the painters’ life experience, the appreciation and comment of their works, the data are divided into eight categories: sad, lonely, pride, wanton, lively, cheerful, quiet and peaceful, forming the emotional data set of TCPs, which includes figure painting, landscape painting, flower and bird painting, animal painting and other contents.

Model training

Training parameters

The five-fold cross validation method was used to verify the performance of the network architecture, that is, the data set was divided into five parts, and the average accuracy of the five results was used to evaluate the model. The parameter settings are shown in Table 1. On the basis of ResNet-50 pre training Imagenet dataset, the new architecture parameters are initialized, and the number of network prediction nodes is set to eight according to the category. Using exponential learning rate decay method, the initial learning rate is 10−3 and the learning rate decay rate is 0.9.

Training results

Compared with VGG-19, GoogLeNet, ResNet-50 and WSCNet, this algorithm has more advantages in TCP emotion recognition task. Each method is fine-tuned under ImageNet pre training, and the experimental results are shown in Fig. 2.

Figure 2 Comparison of different CNN structures.

ResNet-50 has better classification performance than VGG-19 and GoogLeNet. We aggregate the multi-layer channel feature information to calibrate the channel characteristics of the module, which is more suitable for emotional analysis of TCP. Therefore, the average accurancy of the proposed model is better than other algorithms.

Furthermore, the ResNet-50 network structure is categorized into four stages based on the output size of the convolutional layers. These stages, progressing from input to output, consist of 16 residual modules in stages 3, 4, 6, and 13, respectively. Notably, when we traverse from deeper levels to shallower ones, the residual modules within the fourth stage through the first stage undergo optimization to become the Feature Aggregation Recalibration Modules. It is worth noting that when the count of optimized modules is set to 0, it signifies an unaltered ResNet-50 structure. Conversely, when the number of optimized modules is set to 16, which denotes the enhanced network structure is valid.

In line with the experimental outcomes depicted in Fig. 3, it is evident that as the number of optimization modules escalates, there is a progressive augmentation in the accuracy of emotion classification. This observation underscores the efficacy of the proposed Multi-Layer Feature Aggregation Recalibration Module in enhancing the network’s feature extraction capabilities. Consequently, this enhancement significantly bolsters the model’s performance, thereby contributing favorably to the task of recognizing emotions within TCP.

Figure 3 Optimization effect of feature aggregation recalibration module.

Emotion classification results

The confusion matrix of emotion classification accuracy of TCP obtained from the above data set is shown in Fig. 4.

Figure 4 Accuracy of emotion classification in TCP.

Obviously, the classification accuracy rate of “quiet and peaceful” emotion is higher than other categories, because most of the TCPs express the sense of peace and atmosphere by depicting the vast and vast scenery. The common elements such as mountains and rivers and trees that reflect the theme of peace and harmony have strong regularity and stability.

The output convolution layer of the last module of the network (the 49th layer of ResNet-50) is taken as the target convolution layer, so as to realize the visualization of the activation and positioning of the sentimental category of TCP. By calculating the weighted sum of the activation diagrams of four emotion categories, the emotions expressed by TCPs of different themes and techniques can be visually displayed by color. Meanwhile, the regions that contribute most to the expression of emotions in TCPs are highlighted and positioned to obtain multi-category CNN emotion discrimination regions.

Conclusion

To intuitively analyze the emotion of TCP, this article proposes a CNN-based classification model for TCP emotion. Utilizing multi-category weighted activation localization visualization technology, the model generates a “visual interpretation” for the emotional decision-making of the CNN. Experimental results demonstrate that, compared to four other network architectures, the proposed algorithm excels in TCP emotion recognition. The inclusion of a multi-layer feature aggregation recalibration module significantly enhances the features extracted by the network, leading to a marked improvement in model performance. Moreover, the algorithm accurately identifies emotional areas in TCP, minimizing the loss of artistic emotional information. This capability aids in comprehending the artist’s techniques for expressing emotions, facilitating easier learning and appreciation of TCP. The results underscore the model’s effectiveness and its potential contribution to the field of art analysis and education.

Supplemental Information

Supplemental Information 1 Code.

We thank the anonymous reviewers whose comments and suggestions helped to improve the manuscript.

Additional Information and Declarations

Competing Interests

Author Contributions

Data Availability

The authors declare that they have no competing interests.

Xiaofang Du conceived and designed the experiments, performed the experiments, prepared figures and/or tables, and approved the final draft.

Yangfeng Cai conceived and designed the experiments, analyzed the data, performed the computation work, prepared figures and/or tables, authored or reviewed drafts of the article, and approved the final draft.

The following information was supplied regarding data availability:

The code is available in the Supplemental File.

The data is available at Zenodo: Hong, Y. (2022). annotation data for art painting detection and identification (Version v1) [Data set]. Zenodo. https://doi.org/10.5281/zenodo.6551801.

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
