# Peer review of "Design of Chinese painting style classification model based on multi-layer aggregation CNN"

_PeerJ Computer Science, doi:10.7717/peerj-cs.2303_

## Round 0.1 · original submission · Major Revisions

Dear authors
We have received sufficient improvements comments from the reviewers who reviewed your article.
Based on the input received we advise you to update your manuscript in light of the comments and resubmit, particularly focusing on the following points

1. State the superiority of your proposed work over the existing works.
2. Why you choose the multi-layer aggregation CNN, instead of simple CNN?
3. improve the figures quality.

4. Get the language of the manuscript improved from a professional editor.
5. State the constraints and limitations of the work.
Thank you

·

Basic reporting

In the manuscript, the author proposed a new classification method based on CNN and emotional knowledge fusion for the traditional Chinese painting. It extract the emotional feature regions based on image saliency firstly, and then a multi-layer aggregation recalibrated emotional feature module is added to CNN network structure, and finally, the method of multi-class weighted activation location is used to classify TCP in CMYK color space.the method is novel and the experiment is extensive. but there are many type errors in the manuscript. So I advise to polish the paper under the help of the fluent English-speakers. what's more, the author should describe the contributions clearly in the manuscript and express the framework of the method with figure in the former part.

Experimental design

In the manuscript, the author proposed a new classification method based on CNN and emotional knowledge fusion for the traditional Chinese painting. It extract the emotional feature regions based on image saliency firstly, and then a multi-layer aggregation recalibrated emotional feature module is added to CNN network structure, and finally, the method of multi-class weighted activation location is used to classify TCP in CMYK color space.the method is novel and the experiment is extensive. but there are many type errors in the manuscript. So I advise to polish the paper under the help of the native English-speakers. what's more, the author should describe the contributions clearly in the manuscript and express the framework of the method with figure in the former part.

Validity of the findings

Why should we care about the emotional classification of TCP? Please provide further justification about the importance of this research in the introduction. How is this research related to modern psychology or what real-world applications does this research support (the last sentence of the abstract is too vague)?

The parameters used for the analysis must be provided in table.

The architecture of the proposed model must be provided.

Additional comments

Address the accuracy/improvement percentages in the abstract and in the conclusion sections, as well as the significance of these results.

Please conduct a more thorough analysis of the failure cases. For instance, in facial expression recognition, it was experimentally found that angry and surprise are easily confused by CNN-based method. You need to show some example for Cheerful->Lonely as reported in Fig. 7 (0.191 is the largest error).

The equations (1), .........(15) should all be scientifically referred in the main text.

Please do a thorough proofreading of language (grammar and word use). Before this paper can be accepted, its literary presentation needs to be improved.

·

Basic reporting

This paper proposes a classification model of TCP emotion based on CNN by analyzing the different emotional mapping relationships in TCP, in order to deeply explore the artistic value of Traditional Chinese Painting (TCP) and solve the gap between its low-level features and high-level semantics of human emotions. Although the experimental results show the effectiveness of the proposed method, some major concerns still exist, so the reviewer feels that this paper should be further improved carefully before considering a possible publication.

1.There exist some grammar issues throughout the paper including the choices of words, the sentence structure, and the use of articles especially "the", which severely disturbs the readability. A native English speaker is strongly recommended for this task to polish the language and correct the grammar errors.

2.The quality of all figures needs to be improved. Most figures have low resolution.

3.The introduction of the article is not logical enough. The description of some references lacks clear explanation of motivation and contribution.

4.What are the real difficulties that justify author’s work, i.e., what are the most important challenges authors want to handle? Why is it so difficult? I suggest to state
this information clear in the introduction in order to give a better understand of the work.

5.There are no comparisons with SOTA methods to verify the superiorities of your method. This is unacceptable. Please select enough methods to test if your method is effective or not.

6.In the paper, a deep literature review should be further given, particularly regarding the work of discriminative CNN feature learning. Therefore, the reviewer suggests discussing the advances by citing some references, e.g., “Class Attention Network for Image Recognition” (https://www.sciengine.com/SCIS/doi/10.1007/s11432-021-3493-7) and “When deep learning meets metric learning: remote sensing image scene classification via learning discriminative CNNs”.

7.Please provide the full names for all abbreviations when they first appear in the abstract/text.

Experimental design

.

Validity of the findings

.

---

## Round 0.2 · accepted · Accept

Based on the input from the experts on revised version of the paper, I'm pleased to inform you about the acceptance of your manuscript, thank you for your fine contribution.

·

Basic reporting

The research paper titled "Design of Chinese painting style classification model based on multi-layer aggregation CNN" introduces a novel approach to classifying Chinese painting styles using a multi-layer aggregation CNN model. The paper provides a comprehensive overview of the methodology employed, including the preprocessing steps, model architecture, and training process. Experimental results demonstrate the effectiveness of the proposed model in accurately classifying Chinese painting styles, achieving high classification accuracy rates across different painting styles.

Experimental design

The experimental design of the study involved collecting a diverse dataset of Chinese paintings representing various styles and genres. The dataset was preprocessed to extract relevant features and normalize the data for training the classification model. The multi-layer aggregation CNN architecture was then implemented and trained on the dataset using standard machine learning techniques. The model's performance was evaluated using metrics such as accuracy, precision, recall, and F1 score to assess its classification capabilities.

Validity of the findings

The study's findings suggest that the multi-layer aggregation CNN model offers a promising solution for automated Chinese painting style classification, with potential applications in art analysis and curation.

Additional comments

Additional comments may include suggestions for further improving the model's performance or expanding the scope of the study to explore other aspects of Chinese art classification.

·

Basic reporting

The manuscript is well revised and is accepted for publication.

Experimental design

The manuscript is well revised and is accepted for publication.

Validity of the findings

The manuscript is well revised and is accepted for publication.

Additional comments

The manuscript is well revised and is accepted for publication.